# Deep Association between Transglutaminase 1 and Tissue Eosinophil Infiltration Leading to Nasal Polyp Formation and/or Maintenance with Fibrin Polymerization in Chronic Rhinosinusitis with Nasal Polyps

**DOI:** 10.3390/ijms232112955

**Published:** 2022-10-26

**Authors:** Toru Sonoyama, Takashi Ishino, Kota Takemoto, Kensuke Yamato, Takashi Oda, Manabu Nishida, Yuichiro Horibe, Nobuyuki Chikuie, Takashi Kono, Takayuki Taruya, Takao Hamamoto, Tsutomu Ueda, Sachio Takeno

**Affiliations:** Department of Otorhinolaryngology, Head and Neck Surgery, Graduate School of Biomedical Sciences, Hiroshima University, Kasumi 1-2-3, Minami-ku, Hiroshima 734-8551, Japan

**Keywords:** chronic rhinosinusitis, nasal polyp, transglutaminase 1, eosinophil, fibrin polymerization, catalyzation

## Abstract

Transglutaminase (TGM) isoform catalyze the cross-linking reaction of identical or different substrate proteins. Eosinophil has been recognized in chronic rhinosinusitis with nasal polyps (CRSwNP) forming tissue eosinophil in nasal polyp (NP), and TGM isoforms are suggested to be associated with a critical role in asthma and other allergic conditions. The aim of this study was to reveal the association of specific TGM isoform with both the tissue eosinophil infiltration deeply concerning with the intractable severity of CRSwNP and the fibrin polymerization ability of TGM isoform associated with the tissue eosinophil infiltration, which lead to NP formation and/or maintenance in CRSwNP. NP tissues (CRSwNP group) and uncinate process (UP) (control group) were collected from patients with CRSwNP and control subjects. We examined: (1) the expression level of TGM isoforms by using a real-time polymerase chain reaction (PCR) and the comparison to the issue eosinophil count in the CRSwNP group, (2) the location of specific TGM isoform in the mucosal tissue using immunohistochemistry, (3) the inflammatory cell showing the colocalization of specific TGM isoform in Laser Scanning Confocal Microscopy (LSCM) imaging, and (4) the fibrin polymerase activity of specific TGM isoform using sodium dodecyl sulfate polyacrylamide gel electrophoresis (SDS-PAGE). A certain level of TGM 1, 2, 3, 5 expression was present in both the CRSwNP group and the control group. Only TGM 1 expression showed a positive significant correlation with the tissue eosinophil count in the CRSwNP group. The localization of TGM 1 in NP (CRSwNP) laid mainly in a submucosal layer as inflammatory cells and was at the cytoplasm in the tissue eosinophil. Fibrin polymerase activity of TGM 1 showed the same polymerase ability of factor XIIIA. TGM 1 might influence the NP formation and/or maintenance in CRSwNP related to the tissue eosinophil infiltration, which formed fibrin mesh composing NP stroma.

## 1. Introduction

Transglutaminase (TGM) comprises a protein family consisting of eight isozymes that catalyze the cross-linking reaction of identical or different substrate proteins and one isozyme that lacks enzymatic activity in mammals [1]. The catalytic mechanism of eight isozymes is highly conserved with sequence and structural homology [1]. The enzyme catalyzes the formation of the covalent isopeptide bond between the γ-carboxamide groups (–(C=O)NH_2_) of glutamine residue side chains and the ε-amino groups (–NH_2_) of lysine residue side chains or a primary amine (RNH_2_) with the subsequent release of ammonia (NH_3_) [2]. The catalytic reaction is irreversible, and the cross-linked products, often forming high molecular mass, exhibit high resistance to mechanical challenge and proteolytic degradation with the insoluble protein polymer formation [3]. Biological studies indicate that the cross-linking reactions are deeply associated with the physiological functions of the cells, including cell death, cell-matrix interactions of the epidermis and hair, the maintenance of tissue integrity, and so on. The reactions are also recognized as the association of human disease states, such as neurodegenerative diseases and autoimmune conditions, including celiac disease, cancer, and tissue fibrosis [3].

Enzymes of eight isoforms have been reported in the human body, and identified enzymatic isoforms are called factor XIIIA and TGM 1–7, which are independently characterized by their prevalent functions. For example, the factor XIIIA stabilizes the fibrin clots and assists with wound healing [1]. TGM 1 is characterized by cell envelope formation in the differentiation of keratinocytes [1], and TGM 2 is associated with cell death [4], cell differentiation [5], cell matrix stabilization [6], and cell adhesion [6]. As for the TGM isoforms of their specific characteristics, the association between the expression of TGM isoforms in airway epithelium and allergic conditions, including asthma and chronic rhinosinusitis with nasal polyps (CRSwNP), have also been reported. TGM 1 is expressed in normal bronchial epithelium [7]. Pulmonary epithelial cells damaged by allergens trigger TGM 2-mediated interleukin (IL)-33 expression, leading to type 2 responses by recruiting both the innate and the adaptive arms of the immune system [8]. TGM 2 and secreted phospholipase A2 (sPLA2) are expressed at increased levels in asthma, and TGM 2 serves as a regulator of sPLA2-X, which initiates cysteinyl leukotrienes (CysLTs) formation in eosinophils [9]. A steroid-resistant cascade of wingless-type MMTV integration site family member 5a (Wnt5a), TGM 2, and leukotrienes drive late-onset house dust mite-induced allergic airway inflammation [10]. Factor XIII was upregulated in the asthma subjects after allergen exposure, and the expression in the sputum of asthma patients correlated with the type-2 immune response and the airflow limitation [11]. The overproduction of factor XIIIA by M2 macrophages might contribute to the excessive fibrin deposition in the submucosa, which will lead to the formation of nasal polyps (NP) [12]. Therefore, these reports suggested that TGM isoforms lead to the allergic condition through the formation of insoluble protein polymer and the cross-linking modification of proteins. As for asthma and CRSwNP, a variety of TGM isoforms like factor XIIIA may contribute to the pathogenic regulation in intense mucosal edema and NP formation and/or maintenance. Although the variety of investigations about TGM 2 and factor XIIIA in allergic diseases has been demonstrated, no investigation about the whole TGM isoforms (TGM 1–7) has been made into the CRSwNP. The Japanese Epidemiological Survey of Refractory Eosinophilic Chronic Rhinosinusitis (JESREC) study revealed that the disease severity of CRSwNP was deeply associated with both the prevalence of asthma and the severity of eosinophil infiltration in NP [13].

To obtain new insight into the pathogenic regulation in intense mucosal edema and we investigated the expression of TGM isoforms of TGM 1-7 in NP, which may be involved in excessive fibrin deposition and matrix crosslinking reaction such as factor XIIIA, and association between eosinophilic infiltration in NP and the expression levels of TGM isoforms. The results showed that TGM 1 expression was only upregulated in patients with CRSwNP correlated with the extent of tissue eosinophilia and TGM 1 colocalize in the cytoplasm of tissue eosinophil. Additionally, the fibrin polymerization ability of TGM 1 showed the similarity of factor XIIIA in the formation of dimers and polymers of fibrin using thrombin. Thus, TGM 1 produced by tissue eosinophil is a functionally relevant enzyme that is likely involved in the development and/or maintenance of NP with fibrin polymerization without using thrombin. Thus, TGM 1 produced by tissue eosinophil, which is clearly different from factor XIIIA, is a functionally relevant enzyme that is likely involved in the development and/or maintenance of NP with fibrin polymerization.

## 2. Results

### 2.1. TGM Isoform Expression in NP and UP

We enrolled 40 CRSwNP patients (CRSwNP group) and 16 control patients (control group) in the study. The expressions of TGM 1–7 were measured with RT-PCR, and TGM 1, 2, 3, 5 were expressed in most of the samples from both groups, but TGM 4, 6, 7, especially in TGM 6, were not detected in most of the samples from both groups. A significant high expression of TGM 1, 3, 5 and a low expression of TGM 2 were recognized in the CRSwNP group compared to the control group (Figure 1).

### 2.2. TGM Isoform Expression and Tissue Eosinophil Count in NP

The correlation between the tissue eosinophils and the expression of TGM 1, 2, 3, and 5 were investigated in the CRSwNP group. Twenty-three CRSwNP patients who performed tissue eosinophil counts in NP were enrolled in the study. TGM 1 showed only a significant positive correlation (*p* < 0.05) with tissue eosinophil with high degree (*r* = 0.513), but not in TGM 2, 3, and 5 in the Pearson’s correlation coefficient (Figure 2). From linear regression analysis, only TGM 1 expression was predicted by the eosinophil count (*R^2^* = 0.263, *p* < 0.01)

### 2.3. Immunohistochemistry and Laser Scanning Confocal Microscopy (LSCM) Imaging of TGM 1

The immunohistochemical study revealed that inflammatory cells in the submucosal layer in NP (CRSwNP) and epithelial cells and inflammatory cells in the submucosal layer in UP (control) showed clear TGM 1 production, and connective tissue in NP (CRSwNP) and UP (control) weakly showed TGM 1 production. The location of TGM 1 positive inflammatory cells in the submucosal layer in NP (CRSwNP) and UP (control) were similar to the location of the eosinophil in HE stains. The degree of positive cells on TGM 1 in the submucosal layer was suggested to be comparatively higher in NP (CRSwNP) than UP (control) associated with the degree of eosinophil infiltration (Figure 3). LSCM images showed that TGM 1 production was colocalized with MBP^+^ cells, which indicated eosinophil, and the location of TGM 1 was mainly in the cytoplasmic region of MBP^+^ cells (Figure 4).

### 2.4. Fibrin Polymerazation Ability of TGM 1

In the study of the fibrin polymerization ability of TGM 1 in the presence of human α-thrombin, the remaining clot of centrifugation after the removal of supernatants showed that TGM 1 have a fibrin polymerization ability like factor XIIIA (Figure 5A). The supernatants of factor XIIIA and TGM 1 showed a similar polymerization pattern forming the monomers α-, β-, and γ-bands (~50–60 kDa), as well as the cross-linked proteins, including the lighter γ-γ dimers (~117 kDa) and the heavier α-α polymers (>~210 kDa) (Figure 5B).

## 3. Discussion

NP, as one of the main pathophysiology in chronic rhinosinusitis, contains inflammatory cells, such as eosinophil, mast cell, innate lymphoid cells, and neutrophils, which form hyperplastic edematous connective tissue with some seromucous glands and plasma exudation [14]. The main stroma of NP consisted of albumin with excessive fibrin deposition [12,15], and fibrin deposition was caused by factor XIIIA catalyzation forming covalent cross-links between γ-glutamyl and ε-lysyl residues on adjacent fibrin chains and cross-links α2-plasmin inhibitor (α2PI) with fibrin [12].

Factor XIIIA mainly participated in the final stage of coagulation cascade, and factor XIII in plasma transformed into the active form with both thrombin cleavage of activation peptide from factor XIIIA and XIIIB dissociation in the presence of Ca^2+^. However, cellular factor XIII can activate only with the presence of Ca^2+^ compared with plasma factor XIII, and cellular factor XIII derived from M2 macrophage is suggested to mainly contribute to form excessive fibrin deposition in NP [12].

As eight TGM isoform including factor XIIIA contain the high-conserved common catalyzation ability of the cross-linking reaction of identical or different substrate proteins, we assumed the association between other TGM isoform except factor XIIIA and NP formation in CRSwNP. Additionally, as a clinical feature causing a strong tendency for the recurrence of NP after surgery that is deeply associated with the severity of tissue eosinophil infiltration and eosinophilic chronic rhinosinusitis identified with the JESREC criteria features edematous NP and thick mucus production, we also assumed the association between TGM isoform production leading to the fibrin mesh formation and tissue eosinophil infiltration.

The study revealed that NP showed a significant high expression of TGM 1, 3, 5 and a low expression of TGM 2 compared to UP. TGM 2 was previously mentioned regarding the deep association causing an allergic status, including IL-33 expression [8], phospholipase A2 (PLA2) expression in asthma [9], and allergic airway inflammation driven by a steroid-resistant cascade of Wnt5a, TGM 2, and leukotrienes [10], but our results revealed that TGM 2 playing a critical role of both inducer of allergic condition and wound healing was not deeply associated with NP development and/or maintenance in terms of the main protein catalyzation factor. Rather, TGM 1, 3, 5 should be thought of as the main protein catalyzation factors, especially in fibrin formation like factor XIIIA, of NP development and/or maintenance. On the other hand, TGM 1, 3, 5 are mainly known to participate in cornified cell envelope formation in keratinization [16], but they were not previously examined regarding the association of allergic condition.

The significant positive correlation between the level of tissue eosinophils and TGM 1 expression in our results revealed that TGM 1 was suggested to be a main transglutaminase isoform associated with the intractable factor of NP development and/or maintenance under type II inflammation, which is deeply concerning with the ECRS pathophysiology.

TGM isoforms except factor XIIIA were expressed and stored in zymogenic or inactivated forms in a cell, and TGM 1 was stored as the plasma membrane-linkage protein via fatty acyl linkage in the N-terminal cysteine residue and released from the membrane into three 10 + 30 + 60 kDa fragments, which increased its catalytic activity after proteolysis. The 60-kDA part was eventually released from the membrane and acted as a cytosolic enzyme [17], and cleaved TGM 1 was more active than intact TGM 1 [18,19].

In immunohistochemistry, TGM 1 production was recognized mainly in epithelial cells in UP and inflammatory cells in the submucosal layer in UP and Np. From the location of TGM 1 positive inflammatory cells in the submucosal layer in the LSCM image, TGM 1 production was colocalized with MBP^+^ cells, and the location of TGM 1 was mainly in the cytoplasmic region of eosinophil. These results suggest that eosinophil is a main source of TGM 1 in NP, which lead to a significant association between tissue eosinophil and TGM 1 expression in PCR analysis of our data. As TGM 1 was shown as a cytoplasmic enzyme in eosinophil, and cytolysis of eosinophil as eosinophil degranulation was induced by fibrinogen and the excessive activation of type 2 cytokines, such as IL-5, IL-33, platelet-activating factor (PAF), regulated on activation, normal T cell expressed and secreted (RANTES), and tumor necrosis factor (TNF)-alpha [20], cytoplasmic and membrane-linkage TGM 1 may be released via eosinophil degranulation under mucosal edema consisting of exuded plasma protein, including fibrinogen with the excessive activation of type 2 cytokines.

The fibrin polymerization ability of TGM 1 showed a similar polymerization ability as factor XIIIA. These results suggest that both factor XIIIA and TGM 1 are deeply associated with the fibrin clot formation leading to the fibrin mesh in NP. These transglutaminases also catalyze a variety of substrates [21], but currently, the complete list of known protein TGM’s substrate is underdeveloped because of a lack of the classification of enzymatic function [21]. However, the overproduction of factor XIIIA and TGM 1 may accelerate the catalyzation of both fibrin and other broad substrate proteins, resulting in excessive fibrin deposition and polymer protein formation, which in turn retains a variety of exuded plasma proteins leading to intense edema or pseudocyst formation in the submucosa of NP tissue. However, there are some limitations. (1) Our study could not reveal how TGM 1 catalyze the NP stroma under the existence of factor XIIIA, either cooperatively, independently, or exclusively in fibrin clot formation. (2) Our research could not explore the catalysis of these transglutaminases on other substrates in NP. (3) Microregional TGM 1 concentration level, especially in the site of eosinophil degranulation, could not be detected because of the limitation of the measurement technique. Nevertheless, as eosinophil is the major inflammatory cell in nasal polyps and is deeply associated with the intractable severity of CRSwNP, enhancing the type of CRS as ECRS, our results imply that the local production of TGM 1 via eosinophil degranulation might be a therapeutic target for controlling the NP formation and/or maintenance in CRSwNP, especially in ECRS.

## 4. Materials and Methods

### 4.1. Patient and Biopsy Specimens

Patients with CRSwNP and control patients who did not have any CRS, but needed endoscopic surgery, such as the correction of nasal deviation, were recruited at Hiroshima University Hospital. The diagnosis of CRS was based on computed tomography (CT) scanning, patient history, clinical symptoms, and endoscopic findings. The inclusion criteria for CRSwNP were as follows: treatment without oral/nasal steroids within 4 weeks before surgery and no improvement in continuous nasal drip, post-nasal drip, and nasal congestion after medical treatment, including low-dose macrolide therapy. Patients with an established immunodeficiency, pregnancy, coagulation disorder, diagnosis of classic allergic fungal sinusitis, Churg–Strauss syndrome, or cystic fibrosis were excluded from the study. Uncinate process (UP) from control patients and nasal polyps (NP) from CRSwNP patients were obtained from routine endoscopic sinus surgery. The specimens fixed in 4% paraformaldehyde for immunohistochemistry or unfixed for Laser Scanning Confocal Microscopy (LSCM) imaging were embedded into the optimal cutting temperature (OCT) compound and stored in the deep freezer at −80 °C following liquid nitrogen freezing. For reverse transcription-polymerase chain reaction (RT-PCR), the specimens were collected with immersing in RNA later solution (Ambion, Austin, TX, USA).

### 4.2. Real-Time PCR

A quantitative PCR analysis was performed on an ABI Prisms 7300 system (Applied Biosystems, Foster City, CA, USA). Cellular RNA was isolated using RNeasy mini-kits (Qiagen, Valencia, CA, USA). Total RNA was then reverse-transcribed to cDNA using a High-Capacity RNA-to-cDNA kit (Applied Biosystems). Gene expressions were measured via a real-time PCR system using TaqMan Gene Expression Assays (Life Technologies, Carlsbad, CA, USA). PCR primers specific for TGM 1 (Hs00165929_m1), TGM 2 (Hs01096681_m1), TGM 3 (Hs00162752_m1), TGM 4 (Hs00162710_m1), TGM 5 (Hs00909973_m1), TGM 6 (Hs00975389_m1), and TGM 7 (Hs00369497_m1) were used. Primers for GAPDH (Hs03929097_g1) were used as a reference. The PCR cycles were run in triplicate for each sample. Amplifications of the PCR products were quantified by the number of cycles and analyzed using the comparative cycle threshold method (2^−∆∆Ct^). Target gene expressions were presented as relative rates compared to the expression of the reference gene (ratio: target gene/GAPDH expression).

### 4.3. Immunohistochemistry and Laser Scanning Confocal Microscopy (LSCM) Imaging

Cryostat sections (approx. 5 μm-thick) were obtained using a cryostat (Leica, Nussloch, Germany). For immunohistochemistry, each slide section was fixed with 99.5% ethanol and incubated with 0.3% H_2_O_2_ in methanol. After immersing with Blocking One Histo (Nacalai Tesque Inc., Kyoto, Japan) for 60 min, the slides were incubated with anti-human TGM 1 Polyclonal Antibody (PA5-59088) (Rabit)/(Thermo Fisher Scientific, Wilmington, DE, USA) diluted with Blocking One Histo (Nacalai Tesque Inc., Kyoto, Japan) in PBST with Tween20 at 4 °C overnight. The tissue slides were washed with PBS and subsequently incubated with Histofine Simple Stain MAX PO (#424151)/(Nichirei Bioscience Inc., Tokyo, Japan). Sections were rinsed and incubated in DAB reagent (#425011)/(Nichirei Bioscience Inc., Tokyo, Japan), and then they were counterstained with hematoxylin. Consecutive sections were stained with hematoxylin-eosin (H.E) for the assessment of mucosal pathology and the degree of tissue eosinophil infiltration. Tissue eosinophil count of the NPs was calculated by measuring the mean cell count of the 3 most dense areas of eosinophils in H.E-stained sections under ×400 magnification. For Laser Scanning Confocal Microscopy (LSCM) imaging, the sections were washed and permeabilized with phosphate-buffered saline (PBS) containing 0.1% Triton-X (Sigma Aldrich, St. Louis, MO, USA) (PBS-T) for 20 min at room temperature. Non-specific binding was blocked by incubating the tissue sections with Image-iT™ FX Signal Enhancer (#I36933)/(Invitrogen, Carlsbad, CA, USA) for 30 min at room temperature. Tissue sections were then incubated overnight at 4 °C with anti-human TGM 1 Polyclonal Antibody (PA5-59088) (Rabit)/(Thermo Fisher Scientific, Wilmington, DE, USA) (1:2000) and anti-human Eosinophil Major Basic protein (CBL419) (mouse)/(Merck KGaA, Darmstadt, Germany) (1:500), then they were reacted with the appropriate secondary antibodies, including anti-mouse IgG Alexa Fluor^®^ 555 (#A21427) (rabbit)/(Invitrogen, Carlsbad, CA, USA) (1:100) and anti-rabbit IgG Alexa Fluor^®^488 (#A11008) (goat)/(Invitrogen, Carlsbad, CA, USA) (1:100). Nuclear staining was done with DAPI (#71-03-00)/(KPL, Gaitherburg, MD, USA) in PBST with 0.01% sodium azide at room temperature for 10 min. Following this, the sections were washed with PBS and mounted with Dako Fluorescence Mounting Medium (#S3023) (Dako, Glostrup, Denmark). Fluorescence-immunolabeled images were acquired using an LSCM (Fluoview FV-1000 D)/(Olympus, Tokyo, Japan). Control specimens with IgG1 isotype control were used to verify that the nonspecific binding was not detectable.

### 4.4. Fibrin Polymerization Ability

A solution containing 1.75 mg/mL fibrinogen (061-03691)/(Wako Pure Chemical Co., Tokyo, Japan) and 0.5, 1, 2, 4 µg/mL factor XIIIA (HCXIIIA-0165)/(Prolytix, Essex Junction, VT, USA) or 0.5, 1, 2, 4 µg/mL TGM 1 (T009)/(Zedira Gmbh, Darmstadt, Germany) in a 50 mM Tris HCl buffer of pH 7.4 containing 10 mM CaCl_2_ was incubated and then clotted in the presence and absence of 1.25 µg/mL human α-thrombin (Prolytix, Essex Junction, VT, USA). The clots were incubated for 24 h at room temperature before the addition of denaturing buffer of 25 mM NaH_2_PO_4_, 5.7 M urea, 1.9% (*w*/*v*) SDS (2331810)/(Atto, Tokyo, Japan) and 1.9% (*w*/*v*) DTT (Wako Pure Chemical Co., Tokyo, Japan), and then they were incubated overnight at room temperature. Samples were boiled in a water bath for 10 min before centrifugation at 12,000× *g* at 20 °C for 3 min. The supernatants at 0.5 µg/mL factor XIIIA and 0.5 µg/mL TGM 1 were examined by SDS-PAGE on homogeneous 10% cross-linked gels and stained with Coomassie Brilliant Blue (#1610803)/(Bio-Rad Laboratories, Hercules, CA, USA).

### 4.5. Statistical Analysis

All data were reported as mean ± SEM unless otherwise noted. Differences between groups were analyzed with Mann–Whitney U-test for the between-group analyses. Correlations were assessed using the Pearson’s correlation coefficient, and linear regression was added in each graph. A *p* value of less than 0.05 was considered statistically significant. Statistical analyses were performed with GraphPad Prism 8 (GraphPad Software, La Jolla, CA, USA).

## Figures and Tables

**Figure 1 ijms-23-12955-f001:**
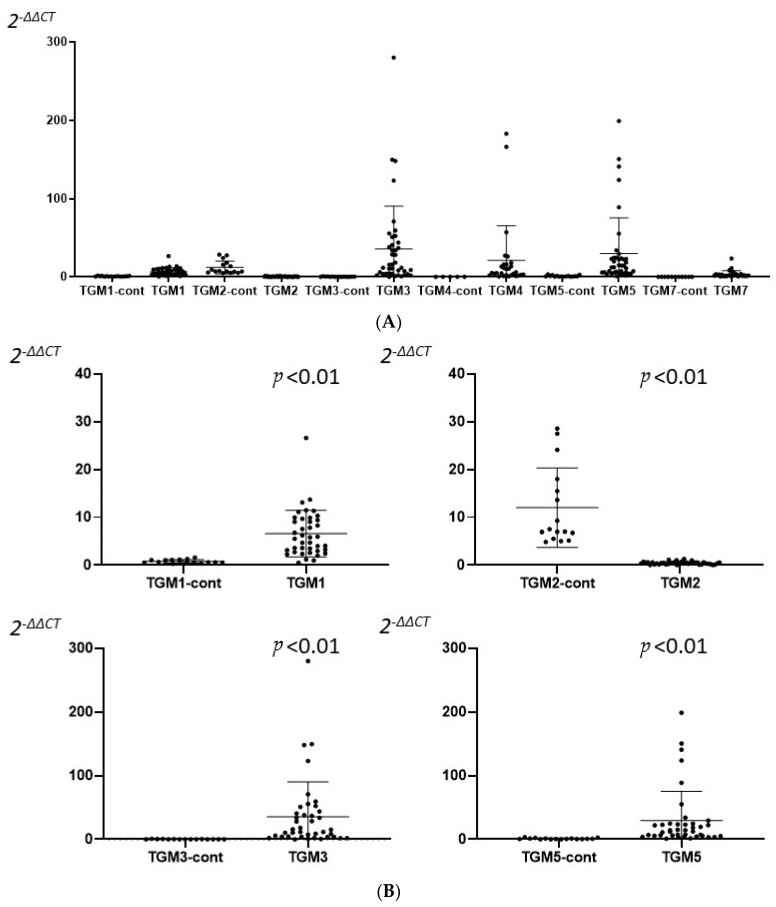
A comparison between the CRSwNP (XX) and the control (XX-cont) group in mRNA expression of each TGM isoform (**A**) and expression of TGM 1, 2, 3, and 5 (**B**). (**A**) TGM 1, 2, 3, 5 isoforms were admitted in both groups. A significant number of TGM 4, 7 isoforms were not admitted. TGM 6 were not admitted at all. (**B**) Expression of TGM 1, 2, 3, and 5. A significant differentiation of TGM 1, 2, 3, and 5 mRNA expression were recognized in the CRSwNP group compared to the control group. TGM 1, 3, and 5 were significantly down-regulated in the control group, but they were significantly up-regulated in the CRSwNP group. Differences between groups were analyzed with Mann–Whitney U-test.

**Figure 2 ijms-23-12955-f002:**
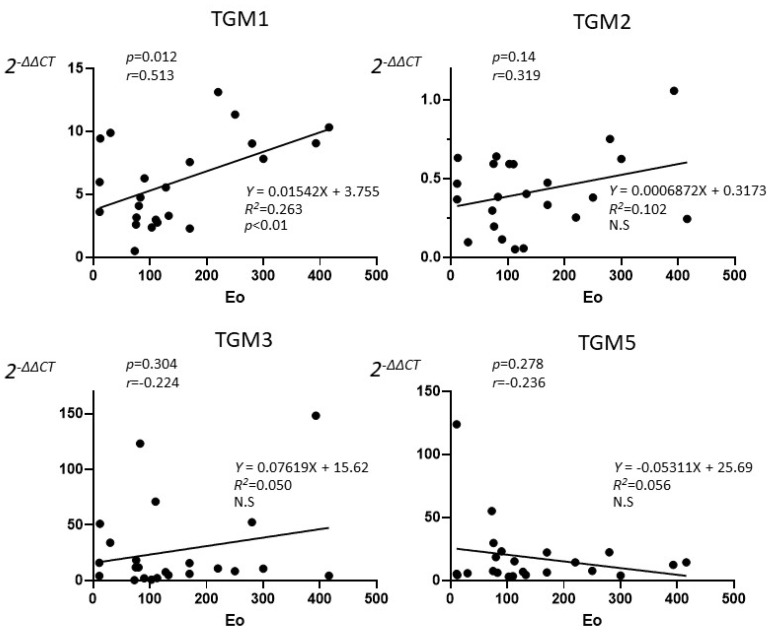
The correlation between the tissue eosinophils counts and the expression of TGM 1, 2, 3, and 5 in the CRSwNP patients who performed tissue eosinophil counts in NP (*n* = 23). Pearson’s correlation coefficient showed only a significant correlation between TGM 1 and tissue eosinophil with high degree (*p* = 0.012, *r* = 0.513), but TGM 2 (*p* = 0.138, *r* = 0.319), TGM 3 (*p* = 0.304, *r* = 0.224), TGM 5 (*p* = 0.278, *r* = −0.236) did not show significance. Each line showed linear regression. From linear regression analysis, only TGM 1 expression was predicted by the eosinophil count (*R^2^* = 0.263, *p* < 0.01).

**Figure 3 ijms-23-12955-f003:**
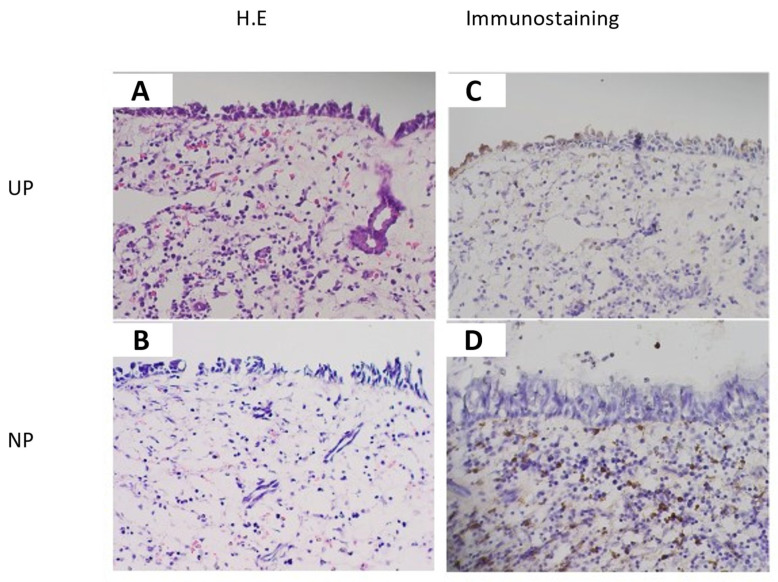
Immunohistochemical images of UP (control) (**A**,**C**) and NP (CRSwNP) (**B**,**D**) (original magnification ×200). TGM 1 production was recognized in epithelial cells and inflammatory cells in the submucosal layer in UP (control) (**C**) and inflammatory cells in the submucosal layer in Np (CRSwNP) (**D**), and it was weakly recognized in the connective tissue in UP (control) (**C**) and NP (CRSwNP) (**D**). The location of TGM 1 positive inflammatory cells in the submucosal layer in UP (control) and NP (CRSwNP) were similar to the location of eosinophil in hematoxylin-eosin (H.E) stains (**A**,**B**). The degree of positive cells on TGM 1 in the submucosal layer was suggested to be comparatively higher in NP than in UP (control) associated with the degree of eosinophil infiltration (**C**,**D**).

**Figure 4 ijms-23-12955-f004:**
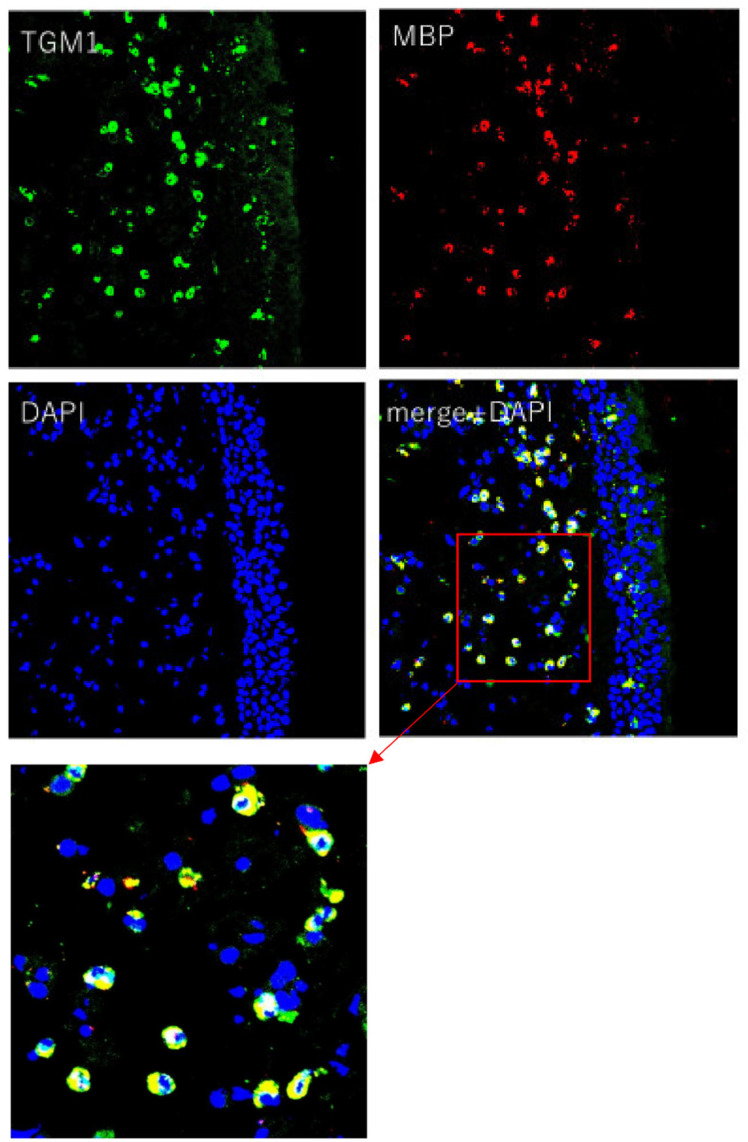
LSCM images of NP (original magnification ×200). LSCM images showed that TGM 1 production was colocalized with MBP (Eosinophil Major Basic protein) positive cell as eosinophils and the location of TGM 1 was mainly in the cytoplasmic region of eosinophil. Immunofluorescence assay was performed with anti-TGM 1 (green fluorescence) and eosinophil (red fluorescence). Nuclei were counterstained with 4′,6-diamidino-2-phenylindole (DAPI; blue fluorescence).

**Figure 5 ijms-23-12955-f005:**
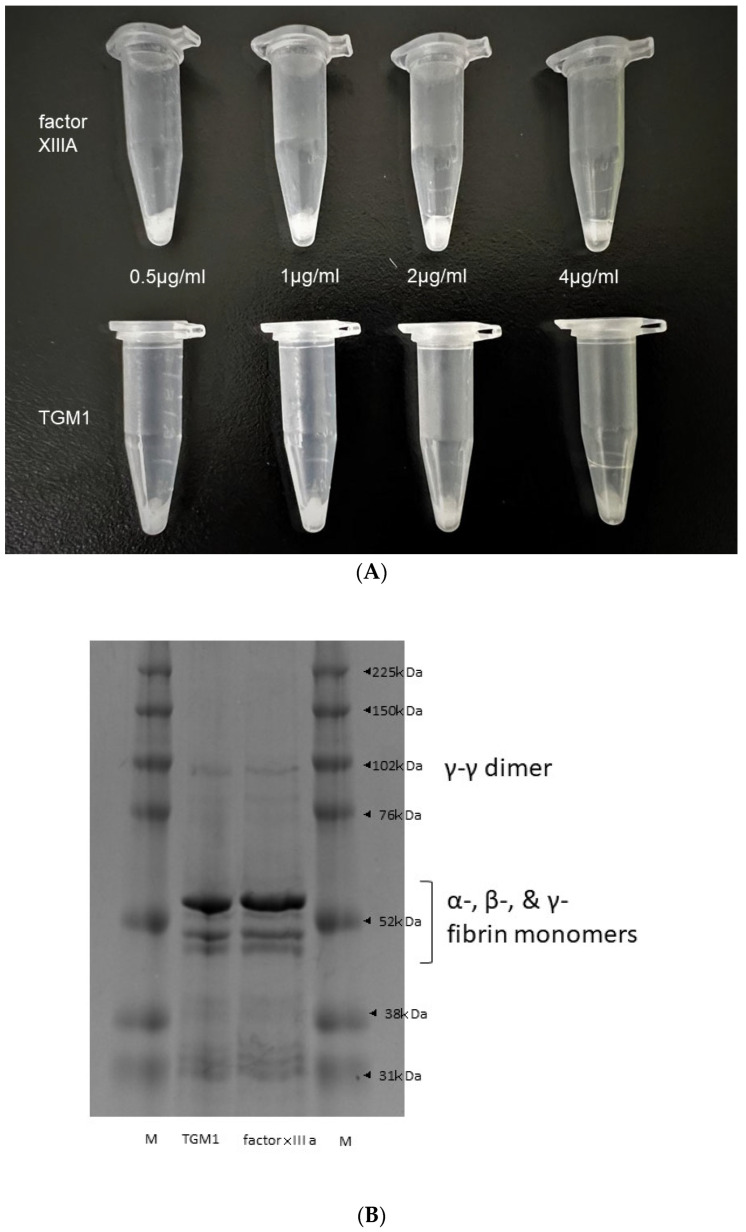
The clotting effect of TGM 1 compared to factor XIIIA. Fibrin clot after the removal of supernatants. (**A**) 0.5, 1, 2, 4 µg/mL factor XIIIA (upper from left to right) and 0.5, 1, 2, 4 µg/mL TGM 1 (lower from left to right) showed a similar formation of fibrin clot by visual inspection. (**B**); SDS-PAGE showed the similarity of the migration pattern of fibrin monomers and polymers obtained using TGM 1 and factor XIIIA.

## Data Availability

Not applicable.

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
