# Peer review of "Deep Association between Transglutaminase 1 and Tissue Eosinophil Infiltration Leading to Nasal Polyp Formation and/or Maintenance with Fibrin Polymerization in Chronic Rhinosinusitis with Nasal Polyps"

_ijms, 2022, doi:10.3390/ijms232112955_

Round 1
Reviewer 1 Report
Paper by Sonoyama et al. presents some data on the role that some TGM isoforms might have a role in NP formation and/or maintenance in CRSwNP, related to the tissue eosinophil infiltration and fibrin deposition leading to NP development.
Paper have some criticisms.
Abstract report a large number of abbreviations not explained. Abstract conclusions should be rewritten after an accurate revision of the paper (see below)
In the introduction text other abbreviation are not explained: CysLT refers to The cysteinyl leukotrienes (CysLTs)? Wnt5a refers to 'Wingless/Integrated' family member 5a?
Results: The continuous exchange of UP instead of control and NP for patients reduces paper readability. It is necessary to uniform the description.
Figure 1. Caption is confounding and should be reshaped
Figure 2. Are the correlation between tissue eosinophils counts and the expression of TGMs obtained using all (patients and controls) data? Considering the number of dots reported it is probable that only some data were considered. Authors should explain their choice.
Figure 5. Data reported in figure 5 are inconclusive . The interpretation of the amount of fibrin production comparing vial containing by visual inspection is almost “subjective”. A turbidimetric method appears more appropriate. TGM1 by SDS-PAGE analysis should be confirmed by immunostaining on a western blot.
In addition, caption reports a sentence that appear to be a probable typo error “LSCM images showed that TGM1 production was colocalized with eosinophils and the location of TGM1 was mainly in cytoplasmic region of eosinophil” when the pro-clotting effect of TGM1 was described.
Discussion line 195-96: Authors claim that TGM2 was not deeply associated with NP development and/or maintenance. Data presented in the paper demonstrated only that the TG is not expressed in submucosal NP, but they have not data on TGM2 role in NP pathogenesis.
Informed Consent Statement: Authors report that Written informed consent was obtained from all subjects involved in the study, but considering that bioptic sapling is invasive, Ethic committee permission statement report might be cited.
In conclusion paper by Sonoyama et al. appears not suitable for publication in the present form. A certain number of additive Data should be obtained to support opinions reported in the discussion.
Author Response
Thank you for reviewing our manuscript.
We revised the manuscript regarding your suggestions.
We answered your questions and suggestions showing below.
Thank you for your meaningful advices.
Abstract report a large number of abbreviations not explained.
>Thank you for your suggestion. We add the abbreviations not explained.
In the introduction text other abbreviation are not explained: CysLT refers to The cysteinyl leukotrienes (CysLTs)? Wnt5a refers to 'Wingless/Integrated' family member 5a?
>Thank you for your suggestion. We add the abbreviations not explained.
Results: The continuous exchange of UP instead of control and NP for patients reduces paper readability. It is necessary to uniform the description.
>Thank you for your suggestion. We changed the word from Up and NP to control and patients to make it clear.
Figure 1. Caption is confounding and should be reshaped
>Thank you for your suggestion. We changed the figure to show clearly.
Figure 2. Are the correlation between tissue eosinophils counts and the expression of TGMs obtained using all (patients and controls) data? Considering the number of dots reported it is probable that only some data were considered. Authors should explain their choice.
>We use only the patient data for analyzing the correlation between tissue eosinophils counts and the expression of TGMs in NP. The correlation between tissue eosinophils counts and the expression of TGMs was analyzed using the data of only the patients with the tissue eosinophil counts in NP. Therefore, we add the sentence above described.
Figure 5. Data reported in figure 5 are inconclusive. The interpretation of the amount of fibrin production comparing vial containing by visual inspection is almost “subjective”. A turbidimetric method appears more appropriate. TGM1 by SDS-PAGE analysis should be confirmed by immunostaining on a western blot.
In addition, caption reports a sentence that appear to be a probable typo error “LSCM images showed that TGM1 production was colocalized with eosinophils and the location of TGM1 was mainly in cytoplasmic region of eosinophil” when the pro-clotting effect of TGM1 was described.
>Thank you for your suggestion. We showed the figure 5 to explain only the ability of fibrin clot formation whether TGM1 have or not. The efficiency of fibrin clot formation in TGM1 should be measured if the microregional concentration of TGM1 and factor XIIIA can be measured, but the microregional concentration of both transglutaminase in nasal polyp cannot be measured because of lack of measurement techniques. Therefore, we did not show the efficiency of the fibrin clot formation but only the ability of fibrin clot formation in the range of TGM1 and factor XIIIa from 0.5 to 4 µg/m. We will measure the efficiency of fibrin clot production of TGM1 and factor XIIIA by turbidimetric method when we will focus on the efficiency of each transglutaminase ability. To make it clear, we revised the word in the sentence from “same” to “similar”.
>Transglutaminase can catalyze the cross-linking reaction of identical or different substrate proteins, but the transglutaminase is released from the cross-linked protein in the catalyzation reaction. Therefore, we measured the formation of fibrin polymerization without using immunostaining.
>In addition, we remove the sentence of “LSCM images showed that TGM1 production was colocalized with eosinophils and the location of TGM1 was mainly in cytoplasmic region of eosinophil” as a typo.
Discussion line 195-96: Authors claim that TGM2 was not deeply associated with NP development and/or maintenance. Data presented in the paper demonstrated only that the TG is not expressed in submucosal NP, but they have not data on TGM2 role in NP pathogenesis.
>Thank you for your suggestion. We add the adequate statement in the sentence adding “in terms of the main protein catalyzation factor “, to make it clear.
Informed Consent Statement: Authors report that Written informed consent was obtained from all subjects involved in the study, but considering that bioptic sampling is invasive, Ethic committee permission statement report might be cited.
>Thank you for your advice. We already mentioned about the ethics committee permission in the paragraph of Institutional Review Board Statement as “The study was conducted in accordance with the Declaration of Helsinki and was approved by the Ethics Committee of Hiroshima University Hospital, Hiroshima University, Hiroshima, Japan (Hi-136; August.11.2014 as date of approval). “.
Reviewer 2 Report
Takashi Ishino and colleagues pinpointed the role of TGM1 and its correlation with tissue eosinophil infiltration in chronic rhinosinusitis with nasal polyps (CRSwNP). The involvement of TGM1 but no other investigated TGM isoforms is interesting. It is also compelling to observe the degree of eosinophil infiltration correlated with TGM1 expression. The results could shed light on the potential mechanism underlying the development of a subtype of CRSwNP, eosinophilic chronic rhinosinusitis. However, a few figures in the article must be revised. Some experiments could be optimised to provide more information. The article, in general, is well-written and informative and could be accepted for publication in the International Journal of Molecular Sciences after major revision.
Major comments
1. The authors should include in the figure legend the method of statistics and the abbreviations mentioned in the figures.
2. Figure 1A showed the expression of all the TGM isoforms in biopsies from the individuals except for Factor XIIIA. Does it also have upregulation in the cohort? In the same figure, the label of y-axis is missing. Is it ∆∆Ct, 2–∆∆Ct or fold-change?
3. TGM6 was not detected in both groups. Was the quantification performed using densitometry of the bands after RT-PCR? If so, the author may show one representative image with all the isoforms including Factor XIIIA. If it was done with real-time PCR, please load the amplicons on the agarose gels afterward to have a representative image of the band.
4. TGM3 and TGM5 expression in the tissues from NP in figure 1B, some data points were cut off. The y-axis must be adjusted. The author may consider using a logarithm for adjustment of y-axis.
5. The results from figure 2 is compelling. It would be better to show r value in the figure instead of R2, as it was stated in the text and the figure legend.
6. In line 132, the authors stated that TGM1 is localised in the cytoplasmic region. However, it is quite difficult to see it in Figure 4. The author may use higher magnification to highlight the colocalisation.
7. Figure 5 showed both factor XIIIA and TGM1 have similar fibrin polymerisation ability. However, it seems to have no dose-dependent effect. Could the authors determine whether there is an impact of dose on fibrin clot formation by using, for instance, a turbidity assay?
Minor comments:
1. There are a couple of typos in the manuscript that I could notice. Figure 5A, the label on the photo should be factor, instead of factror. Line 166, it should be SDS-PAGE instead of SD-PAGE.
2. Some of the abbreviations were not mentioned throughout the whole text. E.g. MBP+.
3. It would be more informative if the authors could provide sequences of the primers used in the study. It is the same for the antibody against MBP. It was not mentioned.
Author Response
Thank you for reveiwing our manuscript. We revised the manuscript regarding your suggestions and advices. Thank you for giving us a variety of meaningful advices.
Major comments
- The authors should include in the figure legend the method of statistics and the abbreviations mentioned in the figures.
>Thank you for your suggestion. We add the abbreviations not explained.
- Figure 1A showed the expression of all the TGM isoforms in biopsies from the individuals except for Factor XIIIA. Does it also have upregulation in the cohort? In the same figure, the label of y-axis is missing. Is it ∆∆Ct, 2–∆∆Ctor fold-change?
>Thank you for your suggestion. We could not analyse the level of expression whether it was upregulated or not, because the cohort expression data of TGM isoforms in NP have been lacked or widely changed in each study.
>We revised the figure adding the label of y-axis and correct the mistakes of the label of y-axis.
- TGM6 was not detected in both groups. Was the quantification performed using densitometry of the bands after RT-PCR? If so, the author may show one representative image with all the isoforms including Factor XIIIA. If it was done with real-time PCR, please load the amplicons on the agarose gels afterward to have a representative image of the band.
>Thank you for your suggestion. We did the quantification only using real-time PCR. The representative image of amplicons on the agarose gels in all transglutaminases is difficult to show because some transglutaminase was widely ranged in the quantification such as TGM3 and TGM5. We could not decide what range of such TGMs were representative. Therefore, we presented the graph of the expression range using real-time PCR. In factor XIIIA, other author already reported the detail including of the expression, which was mentioned as line 72.
Line 72: Overproduction of factor XIIIA by M2 macrophages might contribute to the excessive fibrin deposition in the submucosa, which will lead to the formation of nasal polyps (NP) [12].
Thus, we focused on the other transglutaminase family except factor XIIIA.
- TGM3 and TGM5 expression in the tissues from NP in figure 1B, some data points were cut off. The y-axis must be adjusted. The author may consider using a logarithm for adjustment of y-axis.
>Thank you for your suggestion. We revised the figure adjusting the y-axis because of mistakes.
- The results from figure 2 is compelling. It would be better to show r value in the figure instead of R2, as it was stated in the text and the figure legend.
>Thank you for your suggestion. We revised the figure adding all the data of correlation and linear regression. Additionally, we add a sentence in figure legend.
- In line 132, the authors stated that TGM1 is localised in the cytoplasmic region. However, it is quite difficult to see it in Figure 4. The author may use higher magnification to highlight the colocalization.
> Thank you for your suggestion. We add a high magnified figure to make it clear.
- Figure 5 showed both factor XIIIA and TGM1 have similar fibrin polymerisation ability. However, it seems to have no dose-dependent effect. Could the authors determine whether there is an impact of dose on fibrin clot formation by using, for instance, a turbidity assay?
>We showed the figure 5 to explain only the ability of fibrin clot formation whether TGM1 have or not. The efficiency of fibrin clot formation in TGM1 should be measured if the microregional concentration of TGM1 and factor XIIIA can be measured, but the microregional concentration of both transglutaminase in nasal polyp cannot be measured because of lack of measurement techniques. Therefore, we did not show the efficiency of the fibrin clot formation but only the ability of fibrin clot formation in the range of TGM1 and factor XIIIA from 0.5 to 4 µg/m. We will measure the efficiency of fibrin clot production of TGM1 and factor XIIIA by turbidimetric method when we will focus on the efficiency of each transglutaminase ability. To make it clear, we revised the word in the sentence from “same” to “similar”.
Minor comments:
- There are a couple of typos in the manuscript that I could notice. Figure 5A, the label on the photo should be factor, instead of factror. Line 166, it should be SDS-PAGE instead of SD-PAGE.
>Thank you for your suggestion. We remove the sentence of figure 5 “LSCM images showed that TGM1 production was colocalized with eosinophils and the location of TGM1 was mainly in cytoplasmic region of eosinophil” as a typo.
- Some of the abbreviations were not mentioned throughout the whole text. E.g. MBP+.
>Thank you for your suggestion. We add the abbreviations not explained.
- It would be more informative if the authors could provide sequences of the primers used in the study. It is the same for the antibody against MBP. It was not mentioned.
>Thank you for your suggestion. We already mentioned in the section of method about the type of TaqMan Gene Expression Assays, but the sequences of the primers PCR system in TaqMan Gene Expression Assays were not revealed from the company. Therefore, we could not write the sequences in detail.
Round 2
Reviewer 1 Report
Authors have substantially revised manuscript according to the major part of suggestions.
However two point should be clearly stated to avoid misinterpretation:
Results lines 121-123, the sentence should be reshaped:
The correlation between tissue eosinophils and the expression of TGM1, 2, 3, and 5 were investigated in the CRSwNP group. We enrolled the study with the 23 CRSwNP, patients positive for tissue eosinophil counts in NP.
Similarly the concept should be clearly stated (selection of only 23 patients because they were positive for tissue eosinophil counts in NP) in figure 2 caption
In addition, to avoid ambiguity in the interpretation of data reported in figure 5 I suggest that caption might be changed as follow: Figure 5. Clotting effect of TGM1 compared to factor XIIIA. Fibrin clot after removal of supernatants. (A) 0.5, 1, 2, 4 µg/mL factor XIIIA (upper from left 176 to right) and 0.5, 1, 2, 4 µg/mL TGM1 (lower from left to right) showed similar formation of fibrin clot by visual inspection . (B); SDS-PAGE showed the similarity of migration pattern of fibrin monomers and polymers obtained using TGM1 and factor XIIIA.
Author Response
Thank you for reviewing our manuscript.
Results lines 121-123, the sentence should be reshaped:
The correlation between tissue eosinophils and the expression of TGM1, 2, 3, and 5 were investigated in the CRSwNP group. We enrolled the study with the 23 CRSwNP, patients positive for tissue eosinophil counts in NP.
Similarly the concept should be clearly stated (selection of only 23 patients because they were positive for tissue eosinophil counts in NP) in figure 2 caption
>Thank you for your meaningful suggestion. We checked the sentence and found the English representation. We corrected the sentence and also add the comment in figure legend.
In addition, to avoid ambiguity in the interpretation of data reported in figure 5 I suggest that caption might be changed as follow: Figure 5. Clotting effect of TGM1 compared to factor XIIIA. Fibrin clot after removal of supernatants. (A) 0.5, 1, 2, 4 µg/mL factor XIIIA (upper from left to right) and 0.5, 1, 2, 4 µg/mL TGM1 (lower from left to right) showed similar formation of fibrin clot by visual inspection . (B); SDS-PAGE showed the similarity of migration pattern of fibrin monomers and polymers obtained using TGM1 and factor XIIIA.
>Thank you for giving us the appropriate representation in the legend of figure 5. We changed the legend as you suggested.
Reviewer 2 Report
The authors have addressed all the comments. The manuscript in its current form can be accepted for publication.
Author Response
Thank you for your consideration in the decision of acceptable for the journal. We revised the manuscript in some minor point depended on the comments of reviewer 1.

Round 3
Reviewer 1 Report
Authors have revised manuscript according to the suggestions. So paper appear suitable for pubbication on IJMS